# Hepatokine and Proinflammatory Cytokine Profile in Patients with Carotid Atherosclerosis and Metabolic Dysfunction-Associated Steatotic Liver Disease

**DOI:** 10.3390/biomedicines13040978

**Published:** 2025-04-16

**Authors:** Ana Delfina Cano-Contreras, Maria del Rocio Francisco, Jose Luis Vargas-Basurto, Kevin David González-Gómez, Hector Vivanco-Cid, Karina Guadalupe Hernández-Flores, Peter Grube-Pagola, Federico Bernardo Roesch-Dietlen, Jose Maria Remes-Troche

**Affiliations:** Instituto de Investigaciones Médico-Biológicas, Universidad Veracruzana. C. Agustín de Iturbide, Col. Ricardo Flores Magón, Veracruz 91700, Mexicojose_luis_3011@hotmail.com (J.L.V.-B.);

**Keywords:** fatty liver, atherosclerosis, hepatokines, inflammation

## Abstract

**Background and Aims:** Hepatokines have a regulatory function in adipose tissue inflammation, metabolic dysfunction-associated steatotic liver disease (MASLD), cardiovascular diseases, and atherosclerosis. Our aim was to evaluate the profile of proinflammatory cytokines and hepatokines in patients with MASLD and carotid atherosclerosis (CA). **Methods:** A prospective and basic research study was conducted on patients with MASLD. Clinical data were collected from a detailed medical history. Liver stiffness was measured using transient elastography, and carotid Doppler ultrasound was performed. Levels of basic biochemical parameters, systemic inflammation markers (TNF-α, IL-6, IL-10, IL-18), and hepatokines (FGF21, ANGPTL4, fetuin-A) were determined. The results were analyzed with SPSS v22.0 software. **Results:** Sixty-seven patients with MASLD were included, 72.1% were women, and the mean patient age was 53.9 + 11.3 years. Atherosclerosis was found in 11 patients (16.2%), and carotid intima–media thickness (CIMT) was altered in the right carotid of 13 patients (19.1%), in the left carotid of 19 (27.9%), and bilaterally in 7 (10.3%). Greater age (*p* = 0.001) and high blood pressure (*p* = 0.028) were correlated with atherosclerosis. There were no differences in systemic inflammation markers, and the hepatokines FGF21 and fetuin-A tended to increase in the presence of CIMT and CA alterations, regardless of fibrosis. **Conclusions:** In our population, patients with MASLD had a 16.6% prevalence of CA, and the risk increased with age and a history of high blood pressure. FGF21 tended to increase in patients with MASLD + atherosclerosis, and fetuin-A was correlated with CIMT alterations, suggesting that the combination of these markers could guide us to suspect early endothelial alterations in patients with MASLD.

## 1. Introduction

Metabolic dysfunction-associated steatotic liver disease (MASLD) is a silent clinical entity that entails multiple hepatic and extrahepatic complications. Pharmacological treatment has focused on controlling comorbidities, and currently, only one medication has been approved by the U.S. Food and Drug Administration for the treatment of patients with MASLD and fibrosis (resmetiron). However, the adoption of a hypocaloric Mediterranean diet (MD), characterized by a low saturated fat content and high consumption of vegetable oils, has proven effective in improving fatty acid profiles and reducing hepatic inflammation. This improvement is reflected in a decrease in transaminases after achieving a weight loss of 5% to 7%. Additionally, the MD enhances insulin sensitivity and optimizes lipid profiles, contributing to the prevention of metabolic diseases associated with MASLD, such as diabetes and hypertension. Furthermore, improvements in fibrosis and inflammation have been observed when weight loss exceeds 10%, regardless of physical activity regimens [1,2].

MASLD has been shown to be an independent risk factor for cardiovascular disease (CVD), with an increased risk of high blood pressure, atherosclerosis, arrhythmias, myocardial dysfunction, and venous thrombosis [3]. Increased carotid intima–media thickness (CIMT) is considered an early marker of generalized atherosclerosis and CVDs. The prevalence of atheromatous plaques in patients with MASLD and no history of CVD is around 21% [4]. Atherosclerosis is one of the main cardiovascular complications of MASLD and is most likely the result of different aggressive factors such as endothelial injury, dyslipidemia, and oxidative stress induced by the inflammatory response present in MASLD [5].

An increase in serum cytokines and tumor necrosis factor alpha (TNF-α), as well as interleukin-6 (IL-6) and interleukin-1β (IL-1β) secreted by Kupffer cells in response to the phagocytosis of apoptotic hepatocytes, has been observed in patients with MASLD. In addition, the IL-6 cytokine has been found to be significantly elevated in patients with atherosclerotic plaque rupture and so is considered a potential predictor of plaque vulnerability. C-reactive protein (CRP) is an inflammation marker mainly produced by the liver in response to the proinflammatory cytokines IL-1β and TNF-α [6,7]. On the other hand, hepatokines secreted by the liver have an effect on glucose and lipid metabolism, indicating the interaction between liver metabolism and the cardiovascular system. These molecules include fetuin-A, fibroblast growth factor 21 (FGF21), and the angiopoietin-like proteins (ANGPTLs) [8,9,10].

The above-mentioned information underlines the need to determine proinflammatory cytokine and hepatokine profile alterations in patients presenting with MASLD and carotid atherosclerosis, considering their possibility of being timely CVD risk markers in these patients.

## 2. Material and Methods

### 2.1. Study Participants

A prospective and basic research study was conducted at the Biomedical Sciences Laboratory of the Research Institute of the *Universidad Veracruzana*. Volunteer patients over 18 years of age who presented with hepatic steatosis identified through liver ultrasound and met the diagnostic criteria of MASLD were selected through convenience sampling that was proportional and representative by age group and sex.

The sampling was conducted for convenience, including patients recently diagnosed with MASLD (within the last 6 months) who were seen at the liver disease clinic of our institute from January to July 2024, based on the availability and accessibility of the participants.

#### 2.1.1. Study Design and Intervention

Past medical history, socioeconomic data, and somatometry results were collected using a questionnaire, and 9 patients (13.4%) reported treatment with statins due to a history of dyslipidemia. After the registration of clinical data, peripheral venous blood samples were then drawn for biochemical profiling (complete blood count, blood chemistry, liver function tests, lipid profile, and insulin) and the quantification of systemic inflammation markers (TNF-α, IL-6, IL-10, IL-18) and hepatokines (FGF21, ANGPTL4, fetuin-A). All patients underwent transient elastography with FibroScan^®^ and carotid Doppler ultrasound, both performed within 7 days of obtaining the peripheral venous blood sample, according to the patients’ availability. Liver stiffness measurement (LSM) and the controlled attenuation parameter (CAP) were determined through the FibroScan^®^ 502 Touch device equipped with M and XL probes (Echosens, Paris, France) performed by a single operator trained and certified by the manufacturer. Only evaluations with at least 10 valid measurements with an IQR below 30% were considered. Carotid Doppler ultrasound was carried out in black and white, with a transducer in the transverse position from the origin of the common carotid artery to the distal sections of the internal and external carotids.

A grayscale and Doppler ultrasound using a transverse transducer was conducted to examine the common, internal, and external carotid arteries. A bilateral evaluation was performed. The intima–media complex was measured on the posterior wall of each common carotid artery 1 cm from the carotid bifurcation, with increased carotid intima–media thickness (CIMT) defined as greater than 1.1 mm. If plaques were present, their morphology and percentage of stenosis were assessed. The studies were conducted by two radiologists who specialize in Doppler ultrasound.

The proinflammatory cytokines TNF-α, IL-6, and IL-1β and the hepatokines FGF21, ANGPTL4, and fetuin-A were individually determined through the enzyme-linked immunosorbent assay (ELISA) technique. Determined according to the manufacturer’s (Invitrogen, Waltham, MA, USA) instructions, IL-6 had sensitivity of 2 pg/mL and a detection range of 2–200 pg/mL; IL-10 had a sensitivity of 2 pg/mL and a detection range of 2–300 pg/mL; and IL-18 (R&D, Minneapolis, MN, USA) had a sensitivity of 11.7 pg/mL and a detection range of 11.7–750 pg/mL. TNF-α (Invitrogen) had a sensitivity of 4 pg/mL and a detection range of 4–500 pg/mL; FGF21 (Invitrogen) had a sensitivity of 8.19 pg/mL and a detection range of 8.19–2000 pg/mL; fetuin-A (Invitrogen) had a sensitivity of 0.205 ng/mL and a detection range of 0.205–20 ng/mL; and ANGPTL4 (Invitrogen) had a sensitivity of 27.4 pg/mL and a detection range of 27.4–20,000 pg/mL.

Throughout the study, the analysts interpreting the results were blinded to the clinical background and outcomes of the study participants. This strategy was implemented to minimize the risk of bias in the interpretation of reference standards.

#### 2.1.2. Statistical Analysis

For analyzing the results, the numerical variables were expressed in measures of central tendency and dispersion, according to their distribution. The categorical variables were expressed as frequency and percentage. Data distribution was evaluated through the Kolmogorov–Smirnov test, and the Levene’s test was applied for examining the homogeneity of variance in the numerical variables. The means of the numerical variables were compared using the Student’s *t* test or the Wilcoxon test according to the nature of the data and their distribution. On the other hand, the categorical variables were compared using the chi-squared test or the Fisher’s exact test. Correlations were evaluated utilizing the Pearson or Spearman coefficients, according to the nature of the variables and data distribution. Statistical significance was set at a *p* < 0.05. We performed a complete case analysis in the case of missing data or indeterminate results. The IBS SPSS version 22.0 program was used for carrying out the statistical analysis and creating the figures and tables.

The study was reviewed and authorized by the research and ethics committee of the *Instituto de Investigaciones Médico-Biológicas* of the *Universidad Veracruzana*, folio IIMB-011-2023 (approval date 02.20.2023). All participants signed a letter of informed consent, in which the procedure, risks, and benefits of the study, as well as the use and protection of their personal information, were explained.

## 3. Results

Sixty-seven patients with MASLD were studied. Most of the patients were women (n = 49, 72.1%); the mean patient age was 53.9 ± 11.3 years; weight was 84.9 ± 17.7 kg; and body mass index was 33.2 ± 6.4 kg/m^2^. The majority of the patients presented with grade 1 obesity (29.4%), followed by grade 2 obesity (26.5%), overweight (25%), grade 3 obesity (10.3%), and normal weight (5.9%).

A total of 19.1% of patients (n = 13) were smokers, and 29.4% (n = 20) were alcohol users; the mean use was 0.32 ± 0.65 g/week and was not significant in any of the cases. The most frequently reported comorbidities were diabetes mellitus (n = 17, 25%), high blood pressure (n = 23, 33.8%), and hypercholesterolemia (n = 15, 22.1%). None of the patients had a history of CVD. Upon analyzing the characteristics of the patients with MASLD with and without atherosclerosis, there was a significant difference with respect to age. No significant differences were found in any of the other characteristics, comorbidities or biochemical parameters. Table 1 and Table 2 shows the population characteristics.

Atherosclerosis was found in 11 patients with MASLD (16.2%), and only 1 patient had 1 unilateral partial stricture (1.4%). Right CIMT was altered in 13 patients (19.1%); left CIMT was altered in 19 patients (27.9%); and there was bilateral alteration in 7 patients (10.3%). The flow velocities of the common carotid artery, the internal carotid, and the external carotid (both the right and left) were significantly different in the patients with atherosclerosis compared with those in the patients with no CIMT alterations, showing a marked decrease in arterial flow velocity. The correlation analysis showed that greater age (95% CI r = 0.1697 to 0.5818, *p* = 0.001) and the presence of high blood pressure (95% CI r = 0.030 to 0.4808, *p* = 0.028) were correlated with the presence of atherosclerosis. However, regarding early changes demonstrated by CIMT alterations, the only correlation was with age (95% CI r = 0.1532 to 0.5705, *p* = 0.002), with no differences in the rest of the comorbidities (Figure 1).

The mean CAP in the patients with atherosclerosis was 309 ± 43 db/m, and it was 300 ± 36 db/m in the patients without atherosclerosis (*p* = 0.439). The majority of the patients presented with severe steatosis (S3) (67.6%), 13.2% with moderate steatosis (S2), and 14.7% with mild steatosis (S1). The alterations in the right CIMT (95% CI r = 0.1179 to 0.3588, *p* = 0.306) and left CIMT (95% CI r = 0.2553 to 0.2286, *p* = 0.343) showed no correlation with the severity of intrahepatic fat infiltration determined through CAP. Regarding liver stiffness, the mean was 7.3 ± 4.5 kPa in the patients with atherosclerosis and 6.3 ± 2.3 kPa in the patients without atherosclerosis (*p* = 0.081). The majority of the patients (55.9%) did not present with fibrosis; 23.6% had mild fibrosis (F1–F2); and 17.6% had a risk for advanced fibrosis (F3–F4), with a mean LSM of 6.52 ± 2.8. The mean LSM in the patients with atherosclerosis and without atherosclerosis was 7.3 ± 4.5 kPa and 6.3 ± 2.3 kPa, respectively, with no statistically significant difference (*p* = 0.292). Alterations in the right CIMT (95% CI r = 0.3206 to 0.1602, *p* = 0.497) and the left CIMT (95% CI r = 0.5252, *p* = 0.497) showed no correlation with liver stiffness severity.

### 3.1. Systemic Inflammation Markers

IL-10 was undetectable in all patients, regardless of the presence of atherosclerosis, and IL-18 was positive in only one patient (1.8%) who presented with MASLD, no atherosclerosis, and normal bilateral CIMT. There were no significant differences in TNF-α and IL-6 between patients with and without atherosclerosis (see Table 3). In the correlation analysis, there was no association of TNF-α levels (95% CI r = −0.3691 to 0.1883, *p* = 0.502), IL-6 levels (95% CI r = −0.3964 to 0.1572, *p* = 0.379), or IL-18 levels (95% CI r = −0.2958 to 0.2665, *p* = 0.914) between the patients with and without atherosclerosis. A sub-analysis was carried out to determine the differences in the levels of systemic inflammation markers in patients with altered CIMT, and no statistically significant differences were found. No association was found in the correlation analysis regarding TNF-α levels (95% CI r = −0–0.3671 to 0.1846, *p* = 0.495), IL-6 levels (95% CI r = −3632 to 0.1890, *p* = 0.515), or IL-18 levels (95% CI r = −0.2446 to 0.3114, *p* = 0.803) between the patients with and without altered CIMT.

### 3.2. Hepatokines

ANGPL4 was positive in only one patient (1.8%) with MASLD and no atherosclerosis or altered CIMT. Said patient had no liver fibrosis and presented with moderate steatosis and grade 1 obesity. In patients with and without atherosclerosis, there were no differences in fetuin-A levels, but there were differences in FGF21, although they were not statistically significant (*p* = 0.067) (see Table 3). The correlation analysis showed a positive correlation between FGF21 and the presence of atherosclerosis (95% CI r = 0.0042 to 0.4607, *p* = 0.046). No correlation was found in ANGPTL4 or fetuin-A (95% CI r = −0.2936 to 0.4607, *p* = 0.658 and 95% CI r = −0.05189 to 0.4152, *p* = 0.121, respectively) (Figure 2).

A sub-analysis was carried out to determine the differences in the hepatokine levels of patients with altered CIMT, and no significant differences were found. Fetuin-A was positively correlated with the presence of altered CIMT (95% CI r = 0.05486 to 0.4967, *p* = 0.016) for alterations in both the right branch (95% CI r = −0.02967 to 0.4335, *p* = 0.085) and the left branch (95% CI r = 0.03651 to 0.4857, *p* = 0.025). No correlation was found in ANGPTL4 or FGF21 (95% CI r = −0.3144 to 0.1631, *p* = 0.518 and 95% CI r = −0.1875 to 0.2916, *p* = 0.657, respectively) (Figure 3).

## 4. Discussion and Conclusions

Our results revealed the elevated prevalence of atherosclerosis in patients with MASLD that had no history of CVD, as well as the correlation of atherosclerosis with increased age and high blood pressure. They also showed that the levels of the hepatokines FGF21 and fetuin-A had a tendency to rise in the presence of altered CIMT and atherosclerosis in patients with MASLD, regardless of fibrosis grade.

Recent studies have shown that patients with MASLD have a higher incidence of cardiovascular events, with the presence of carotid atherosclerosis acting as an early marker of generalized atherosclerosis and, as a result, increasing major cardiovascular events such as ischemic heart disease, stroke, congestive heart failure, and cardiovascular mortality [11,12]. Even though the presence of plaque is a good risk predictor, altered CIMT has been reported to also predict cardiovascular risk [13]. The presence of atheromas in the middle-aged population with mild-to-moderate cardiovascular risk is 41% in the carotid bifurcation and 22% in the internal carotid; one-fifth of those patients can have generalized atherosclerosis [14]. On the other hand, the presence of atheromatous plaques in patients with MASLD and no history of CVD is around 21%, leading to endothelial injury, dyslipidemia, and oxidative stress [15]. Our results showed a similar frequency, given that 16.6% of the population presented with carotid atherosclerosis.

The increase in total cholesterol and especially low-density lipoprotein (LDL) cholesterol is considered a risk factor for the development of atherosclerotic CVD. Yu Z et al. evaluated the presence of carotid plaque in 4821 patients over 60 years of age with high blood pressure and found a positive association between serum cholesterol levels and the risk for carotid plaque, regardless of LDL cholesterol [16]. This was not seen in our patients given that the levels of cholesterol and their fractions or triglycerides were not correlated with altered CIMT or with the presence of atheromatous plaques. This could be due to the presence of other cardiometabolic risk factors that conditioned low-grade inflammation and the production of atherosclerotic plaque, such as age, obesity, and high blood pressure.

The presence of liver fibrosis in patients with MASLD has been associated with an increase in CVD. Schonmann Y et al. carried out an analysis of 8511 patients with MASLD, 195 of whom had a risk of advanced fibrosis, through FIB-4. Their analysis showed that those patients had a higher risk of CVD regardless of their use of statins or their sociodemographic characteristics (HR 1.63 (1.29–2.06) 95% CI), suggesting that fibrosis scores are associated with CVD [17].

In our population, we found no correlation between fibrosis estimated by liver stiffness and the presence of altered CIMT or atheromatous plaques, which may be due to the fact that the majority of our patients did not present with fibrosis or presented with mild fibrosis grades (F1–F2). Thus, increasing the sample size and including a larger number of patients with advanced fibrosis (F3–F4) is recommended to be able to generalize the results to that population.

In MASLD, we are confronted with atherogenic dyslipidemia characterized by elevated serum triglycerides and low high-density lipoprotein levels, added to the predominance of LDL cholesterol and the accumulation of triglyceride-rich lipoproteins that contain the C3 apolipoprotein that activates TLR 2 and 4, leading to the activation of the NLRP3 inflammasome. Caspase-1 activation through the NOD-like receptor protein 3 (NLRP3) inflammasome leads to the proteolytic activation of proinflammatory cytokines of the IL-1β family and the later induction of the IL-1, IL-6, and CRP pathways. These inflammatory mediators in MASLD are also involved in the development of vascular inflammation and atherosclerotic CVD [18,19]. Our results are contradictory, given that an association of inflammatory factors with greater vascular alterations was expected. This discrepancy could be attributed to having excluded patients with advanced disease, i.e., those with CVD and uncontrolled underlying metabolic disorders.

More than twenty hepatokines have been studied, but those that have shown regulatory functions with respect to inflammation in adipose tissue, as well as a relation with CVD and early atherosclerosis, are fetuin-A, ANGPTL1, and FGF21. Fetuin-A is an inhibitor of insulin signaling at the tyrosine kinase level and is associated with insulin resistance. Its expression in liver tissue is positively regulated by free fatty acids via NF-KB signaling and by glucose via ERK1/2 signaling. Different studies describe the association between fetuin-A levels and the risk of CVD conditioned by insulin resistance. However, results have been contradictory, which may be associated with the genetic variations involved in fetuin-A expression [20,21,22]. We found no difference between fetuin-A levels in patients with and without atherosclerosis. Nevertheless, there was a positive correlation with altered CIMT, and so it could be evaluated as a marker of early vascular injury. With respect to ANGPTL4, its circadian expression has been seen in adipose tissue. ANGPTL 3, 4, and 8 appear to be designed for storing fatty acids, directing them to skeletal muscle for later energy provision, which is why they are altered in patients with MASLD who also have dyslipidemia [23]. In our cohort, only one patient was positive for ANGPTL4, with no correlation with clinical parameters, most likely because the levels of cholesterol and its fractions were slightly elevated, with the majority within the normal range. Even though the action of FGF21 in MASLD is not fully understood, it has been shown to interact with TLR4 in liver macrophages, together regulating inflammatory signaling pathways and the NLRP3 inflammasome, conditioning liver injury and the altered lipid metabolism characteristic of MASLD [24,25]. In our patients, FGF21 was higher in MASLD and atherosclerosis, with a positive correlation. Elevated levels of FGF21 have been reported in patients with obesity, insulin resistance, and diabetes mellitus, resulting from elevated hepatic lipid and carbohydrate signaling, with the risk factors of MASLD coinciding with those of atherosclerosis, explaining the positive correlation of that hepatokine [26,27].

Our study limitations are related to sample size, and the analysis of a larger number of patients is needed to generalize results. In addition, the majority of our patients did not present with fibrosis or had mild fibrosis, making it impossible to carry out risk analyses and correlations of atherosclerosis with the presence of fibrosis, as reported in previous studies.

Our results enable us to conclude that in our population, patients with MASLD had a 16.6% prevalence of carotid atherosclerosis, and the risk increased with age and a history of high blood pressure. Early changes represented by altered CIMT were correlated with an increase in age, regardless of comorbidities. On the other hand, given the severity of fat infiltration and fibrosis, MASLD severity was not correlated with the risk for atherosclerosis or altered CIMT.

There was no correlation between systemic inflammation markers and the presence of atherosclerosis and altered CIMT. The hepatokine FGF21 tended to be elevated in patients with MASLD and atherosclerosis, resulting in a positive correlation, and fetuin-A was correlated with altered CIMT, signifying that the combination of those markers could guide us toward suspecting early endothelial alterations in patients with MASLD.

## Figures and Tables

**Figure 1 biomedicines-13-00978-f001:**
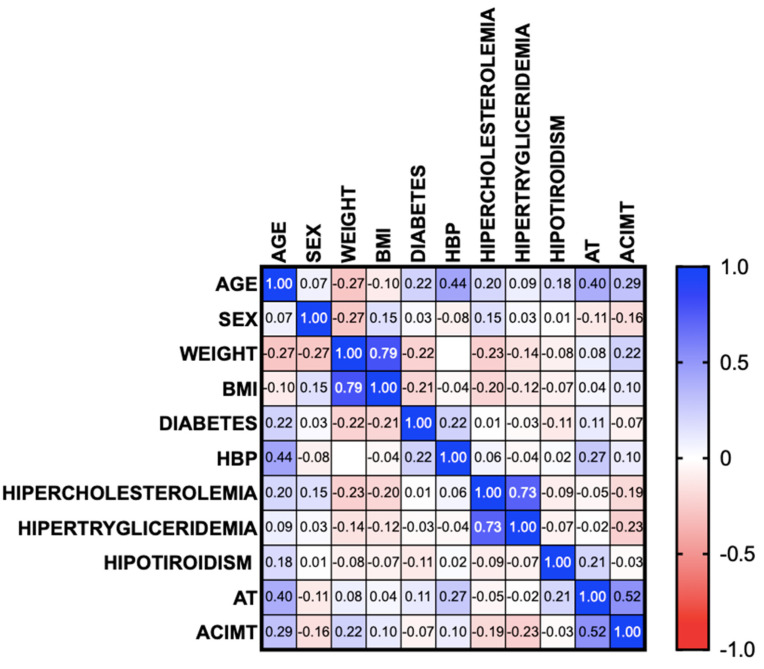
Correlation of the presence of atherosclerosis and altered carotid intima–media thickness with the presence of comorbidities in patients with MASLD. AT: atherosclerosis, ACIMT: altered carotid intima–media thickness, MASLD: metabolic dysfunction-associated steatotic liver disease, BMI: body mass index, HBP: high blood pressure.

**Figure 2 biomedicines-13-00978-f002:**
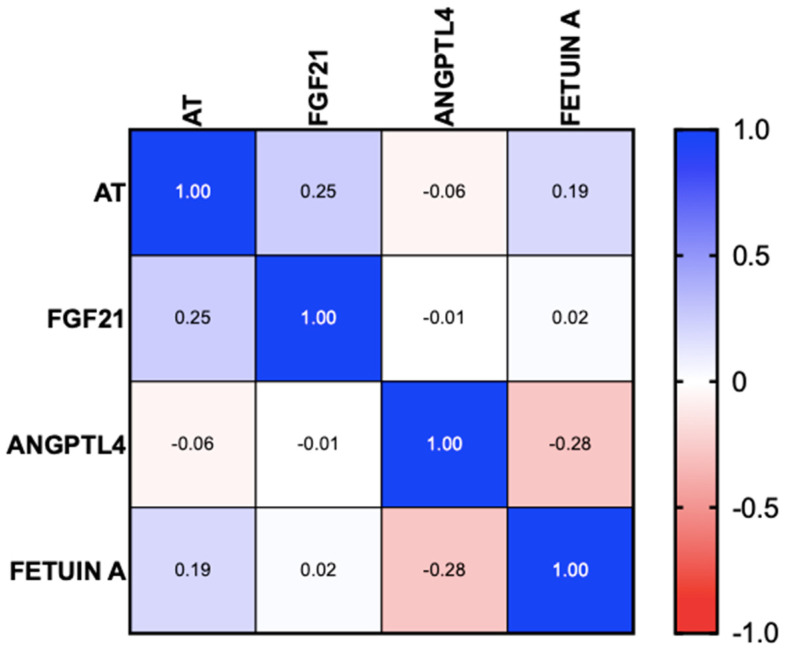
Correlation of hepatokines with the presence of atherosclerosis in patients with MASLD. AT: atherosclerosis, MASLD: metabolic dysfunction-associated steatotic liver disease.

**Figure 3 biomedicines-13-00978-f003:**
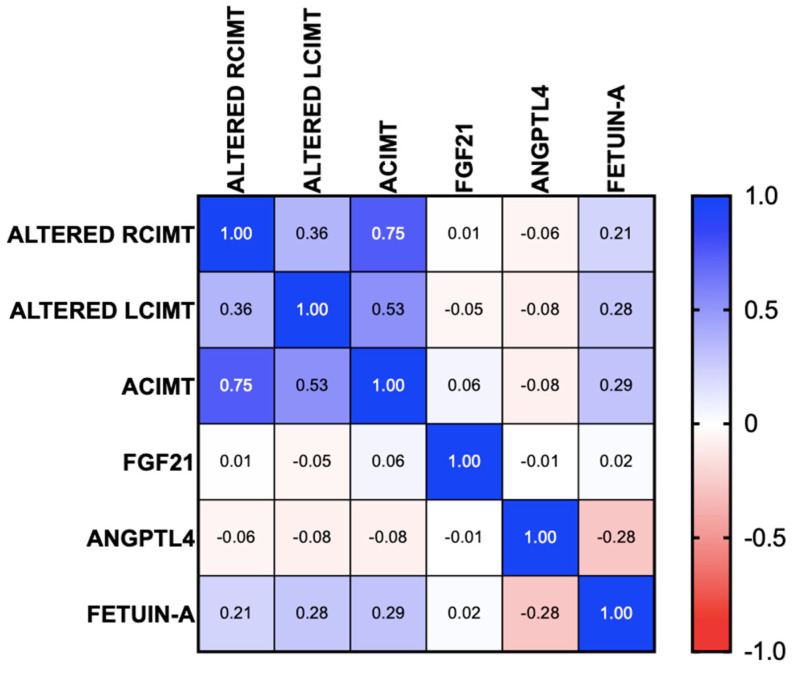
Correlation of hepatocytes with alterations in the carotid intima–media thickness in patients with MASLD. AT: atherosclerosis, MASLD: metabolic dysfunction-associated steatotic liver disease, ACIMT: altered carotid intima–media thickness, RCIMT; right carotid intima–media thickness, LCIMT: left carotid intima–media thickness.

**Table 1 biomedicines-13-00978-t001:** Characteristics of patients with MASLD with and without atherosclerosis.

Parameter	MASLD (n = 67)	MASLD with AT (n = 11)	MASLD Without AT (n = 56)	* p *
Sex				
Male n(%)	17 (25)	4 (36.4)	13 (23.6)	0.193
Female n(%)	49 (72)	7 (63.6)	42 (76.4)	
Age	55.9 +11.3	63 + 5.4	51 + 11.2	0.001
Weight	84.9 + 17.7	88.2 + 11.9	84.3 +18.7	0.379
BMI	33.2 + 6.4	33.8 + 3.7	33.1 + 6.8	0.610
BMI				
Normal weight n(%)	3 (4.4)	-	3 (5.5)	-
Overweight n(%)	17 (25)	1 (9.1)	16 (29.1)	0.086
Obesity I n(%)	20 (29)	4 (36.4)	16 (29.1)	0.157
Obesity II n(%)	19 (27)	6 (54.5)	13 (23.6)	0.086
Obesity III n(%)	7 (10)	-	7 (12.7)	-
Comorbidities				
Diabetes n(%)	17 (25)	4 (36.4)	13 (23.6)	0.386
High blood pressure n(%)	23 (33.8)	7 (63.6)	16 (29.1)	0.028
Hypercholesterolemia n(%)	15 (22.1)	2 (18.2)	13 (23.6)	0.692
Hypertriglyceridemia n(%)	13 (19.1)	2 (18.2)	11 (20)	0.894
Hypothyroidism n(%)	8 (11.8)	3 (27.3)	5 (9.1)	0.094

MASLD: metabolic dysfunction-associated steatotic liver disease, AT: atherosclerosis.

**Table 2 biomedicines-13-00978-t002:** Biochemical parameters of patients with MASLD.

Parameter	MASLD (n = 67)	MASLD with AT (n = 11)	MASLD Without AT (n = 56)	* p *
Hb (g/dL) **	14.3 ± 1.5	13.2 ± 1.8	13.5 ± 1.4	0.463
Platelets **	270 ± 81	221 ± 65	270 ± 81	0.038
TB (g/dL) *	0.45 (0.2–1.70)	0.48 (0.27–1.70)	0.45 (0.2–1.23)	0.136
AST (g/dL) *	24 (12–233)	21 (15–148)	24 (12–233)	0.535
ALT (g/dL) *	20 (5–402)	28 (12–137)	29 (5–402)	0.895
ALP (g/dL) *	97 (10–244)	223 (13–236)	99 (10–244)	0.561
GGT (g/dL) *	35 (3–370)	70 (14–225)	35 (3–370)	0.069
TP (g/dL) *	7.5 (6.3–8.8)	7.5 (6.3–8.8)	7.3 (6.7–7.4)	0.639
ALB (g/dL)	4.2 (6.3–7.4)	4.1 (3.7–4.7)	4.3 (3.7–5.3)	0.042
Cholesterol (g/dL) **	188 ± 51	212 ± 45	185 ± 51	0.114
HDL (g/dL) *	44 (10–214)	53 (10–62)	43 (10–175)	0.837
LDL (g/dL) **	117 ± 46	136 ± 49	115 ± 43	0.215
VLDL (g/dL) *	30 (13–115)	31.8 (19–51)	28 (13–115)	0.215
Triglycerides (g/dL) *	150 (45–576)	158 (99–256)	147 (45–576)	0.936
Glucose (g/dL) *	102 (81–200)	106 (81–155)	101 (83–200)	0.497
Creatinine (g/dL) *	0.71 (0.4–1.45)	0.96 (0.66–1.45)	0.70 (0.4–1.03)	0.009
CRP (g/dL)*	3.65 (1–12)	3.4 (2–7)	3.5 (1–12)	0.568
Insulin (g/dL) **	20.8 ± 10.9	23.7 ± 12	20.5 ± 10	0.618
HOMA *	4.2 (2–6–15.38)	4.1 (3.73–14.18)	4.3 (2.6–15.3)	0.590

MASLD: metabolic dysfunction-associated steatotic liver disease, AT: atherosclerosis, TB: total bilirubin, AST: aspartate aminotransferase, ALT: alanine aminotransferase, ALP: alkaline phosphatase, GGT: gamma-glutamyl transpeptidase, TP: total protein, ALB: albumin, HDL: high-density lipoprotein, LDL: low-density lipoprotein, VLDL: very-low-density lipoprotein. * Non-normal distribution, ** normal distribution.

**Table 3 biomedicines-13-00978-t003:** Systemic inflammation markers in patients with MASLD.

Parameter	MASLD (n = 67)	MASLD with AT (n = 11)	MASLD Without AT (n = 56)	* p *
Systemic inflammation markers				
TNF-α (pg/mL) *	0 (0–125)	0	(0–125)	-
IL-6 (pg/mL) *	0 (0–50)	0	(0–13)	-
IL-10 (pg/mL) *	-	-	-	-
IL-18 (pg/mL) **	819 ± 237	824 ± 210	824 ± 245	NS
Hepatokines				
FGF21 (pg/mL) *	231 (26–2035)	301 (98–2035)	230 (26–1419)	0.067
ANGPL4 (pg/mL) *	8489	-	8489	-
Fetuin-A (ng/mL) *	18.8 (0–19.2)	18.9 (18.5–19.0)	18.8 (0–19.2)	0.796

Note: MASLD: metabolic dysfunction-associated steatotic liver disease, IL: interleukin, NS: not significant, ANGPTL: angiopoietin-like protein, FGF21: fibroblast growth factor 21. * Non-normal distribution, ** normal distribution.

## Data Availability

Patient information is confidential. If needed, data can be requested from the corresponding author upon reasonable request.

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
