# Peer review of "Hepatokine and Proinflammatory Cytokine Profile in Patients with Carotid Atherosclerosis and Metabolic Dysfunction-Associated Steatotic Liver Disease"

_biomedicines, 2025, doi:10.3390/biomedicines13040978_

Round 1

Reviewer 1 Report

Comments and Suggestions for Authors

The work by Cano-Contreras et al. focuses on the profile of proinflammatory cytokines and hepatokines in patients with MASLD and subclinical carotid atherosclerosis. This study included 67 patients who were selected through convenience sampling based on the presence of steatotic liver disease detected by ultrasound and meeting the definition of MASLD (cardiometabolic risk factor). Given that the study population was not biopsy-proven, the impact of hepatic inflammation (steatohepatitis or NAS score) could not be evaluated. However, an assessment of AST and ALT levels above the upper limit of normal, as well as the use of FIB-4 cut-off values, should be considered. The authors observed no association between inflammatory markers or hepatokines with subclinical carotid atherosclerosis in MASLD subjects. They pointed out that elevated prevalence of atherosclerosis in patients with MASLD that had no history of CVD, as well as the correlation of atherosclerosis with older age and hypertension. Based on the only 67 patients sample not significant differences were observed between MASLD subgroup population, this is based on the powerless of the sample. While the study provides insights into MASLD and carotid atherosclerosis, novelty is scarce and should be better emphasized in the manuscript. Nevertheless, several critical issues need to be addressed to improve the overall quality of this work.

Major comments:

1) The authors describe a sample size formula that is incorrect for the study design. Please revise this, as it is a critical issue. The sample size is clearly underpowered. The study only includes 67 patients, which is relatively small for meaningful subgroup analysis. It is recommended that the authors consult a statistical expert for the sample size calculation and statistical analysis for subgroup evaluation.

2) A thorough English language polishing is required to enhance readability all over the manuscript.

Minor comments:

Abstract:

1) This is a clear cross-sectional study, so causality cannot be established. “An observational, prospective, cross-sectional, basic research study was conducted” this phrase is overly complex and should be simplified.

2) Please consider rephrasing this paragraph “Clinical history, somatometry, hepatic elastography, and carotid Doppler ultrasound were performed” for “ Clinical data were collected from a detailed medical history. Liver stiffness was measured using transient elastography, and carotid Doppler ultrasound was performed.”

Introduction:

3) Please add the appropriate reference of this paragraph: “The prevalence of atheromatous plaques in patients with MASLD and no history of CVD is around 21%”.

Material and Methods:

4) The authors should clearly described the time lapse between the diagnosis of MASLD, biochemical sample analysis, liver stiffness measurements, and carotid Doppler ultrasound.

5) The authors should clearly stated the definition of carotid atherosclerosis by doppler ultrasound.

6) Statin or fibrates used should be clearly documented in clinical evaluation.

Results:

7)  IL-10 was undetectable in all patients? It is important that the authors should address this technical issue.

References:

8) References [2] and [9] are duplicated. Please, ensure proper order formatting all over the manuscript.

Comments on the Quality of English Language

A thorough English language polishing is required to enhance readability all over the manuscript.

Author Response

Major comments:

1) The authors describe a sample size formula that is incorrect for the study design. Please revise this, as it is a critical issue. The sample size is clearly underpowered. The study only includes 67 patients, which is relatively small for meaningful subgroup analysis. It is recommended that the authors consult a statistical expert for the sample size calculation and statistical analysis for subgroup evaluation.

Response: After reviewing the sample calculation again, we confirm the validity of your observation. Therefore, we have modified and clarified the period during which the study was conducted, using convenience sampling. Although the sample is relatively small, this is a pilot study that highlights the importance of this type of analysis in our centers.

2) A thorough English language polishing is required to enhance readability all over the manuscript.

Response: Language translation and correction were performed by an expert translator.

Minor comments:

Abstract:

1) This is a clear cross-sectional study, so causality cannot be established. “An observational, prospective, cross-sectional, basic research study was conducted” this phrase is overly complex and should be simplified.

2) Please consider rephrasing this paragraph “Clinical history, somatometry, hepatic elastography, and carotid Doppler ultrasound were performed” for “ Clinical data were collected from a detailed medical history. Liver stiffness was measured using transient elastography, and carotid Doppler ultrasound was performed.”

Response: Abstract: The phrase “observational, prospective, cross-sectional basic research study” was simplified to “observational research,” and modifications were made to the methodology wording.

Introduction:

3) Please add the appropriate reference of this paragraph: “The prevalence of atheromatous plaques in patients with MASLD and no history of CVD is around 21%”.

 Response: Introduction: A reference was added

Material and Methods:

4) The authors should clearly described the time lapse between the diagnosis of MASLD, biochemical sample analysis, liver stiffness measurements, and carotid Doppler ultrasound.

5) The authors should clearly stated the definition of carotid atherosclerosis by doppler ultrasound.

6) Statin or fibrates used should be clearly documented in clinical evaluation.

Response: Materials and Methods: The time between the collection of peripheral venous blood samples, liver elastography, and carotid ultrasound is described; a more detailed explanation of the carotid Doppler ultrasound study was provided; the number and percentage of patients on statin treatment before evaluation are mentioned. 

Results:

7)  IL-10 was undetectable in all patients? It is important that the authors should address this technical issue.

Response: In results: IL-10 was indeed undetectable in all patients; a second analysis was conducted to corroborate our results, which were the same.

References:

8) References [2] and [9] are duplicated. Please, ensure proper order formatting all over the manuscript.

Response: References were reviewed and reference 9 was modified.

Reviewer 2 Report

Comments and Suggestions for Authors

This original work conducted an analysis of systemic inflammatory markers and hepatokines in the blood of patients with MASLD, relating these levels to carotid atherosclerosis.

Some suggestions:

  1. I would like the authors to comment on the observational nature of this study: is the measurement of all these markers routinely performed in clinical practice in your country? Are there local guidelines that support this?
  2. In the materials and methods section, the study is described as both prospective and cross-sectional. This needs to be clarified, as these terms are oxymoronic when used together;
  3. In the abstract, it is unnecessary to mention that SPSS was used to analyse the data;
  4. I would divide the methods into different sections to facilitate the reader’s identification of the relevant points and provide more detail on the inclusion and exclusion criteria;
  5. In the introduction, I would mention the current therapeutic approaches for MASLD, also citing non-pharmacological therapies such as physical activity and the modulation of drinking water, as in the case of Fonte Essenziale water;
  6. I believe the manuscript would benefit from figures illustrating all the variations in the molecular variables assessed by the authors.

Author Response

Comments 1: I would like the authors to comment on the observational nature of this study: is the measurement of all these markers routinely performed in clinical practice in your country? Are there local guidelines that support this?

Response 1: Our study is exploratory; these markers are not measured in daily clinical practice due to high costs.

Comments 2: In the materials and methods section, the study is described as both prospective and cross-sectional. This needs to be clarified, as these terms are oxymoronic when used together.

Response 2: Corrections were made to the methods section.

Comments 3: In the abstract, it is unnecessary to mention that SPSS was used to analyse the data.

Response 3: SPSS analysis was removed from the abstract.

Comments 4: I would divide the methods into different sections to facilitate the reader’s identification of the relevant points and provide more detail on the inclusion and exclusion criteria.

Response 4: Methods were divided into different sections.

Comments 5 & 6: In the introduction, I would mention the current therapeutic approaches for MASLD, also citing non-pharmacological therapies such as physical activity and the modulation of drinking water, as in the case of Fonte Essenziale water. I believe the manuscript would benefit from figures illustrating all the variations in the molecular variables assessed by the authors.

Response 5 & 6: We do not have images illustrating the evaluated molecular variables

Round 2

Reviewer 1 Report

Comments and Suggestions for Authors

Dear Authors,

Thank you for providing the revised manuscript. I have carefully reviewed this updated version and can confirm that you have addressed all of my comments and suggestions. Based on the comprehensive revisions and clarifications you have provided, I believe your manuscript now meets the necessary criteria for publication.

Author Response

Thank you for your comments and contributions to our work, they are very useful to improve.

Reviewer 2 Report

Comments and Suggestions for Authors

The authors have not provided sufficient commentary on the matter. If you assert that your biomarkers are not measured and assessed in standard clinical practice, then the term "observational" must be removed from every part of the manuscript. 

Recommendations were provided for integrating the introduction (e.g. "In the introduction, I would mention the current therapeutic approaches for MASLD, also citing non-pharmacological therapies such as physical activity and the modulation of drinking water, as in the case of Fonte Essenziale water"), but they were neither followed nor addressed.

Comments on the Quality of English Language

English should be improved.

Author Response

The authors have not provided sufficient commentary on the matter. If you assert that your biomarkers are not measured and assessed in standard clinical practice, then the term "observational" must be removed from every part of the manuscript. 

Our study is exploratory; there is limited evidence regarding the clinical impact of this set of biomarkers in daily clinical practice. This is why it was our study objective, seeking their presence in our population. Thank you for your observation; we have modified the terms, and it is defined as a prospective and basic research study.

Recommendations were provided for integrating the introduction (e.g. "In the introduction, I would mention the current therapeutic approaches for MASLD, also citing non-pharmacological therapies such as physical activity and the modulation of drinking water, as in the case of Fonte Essenziale water"), but they were neither followed nor addressed.

We made the suggested changes in the introduction.

Language translation and correction were performed by an expert translator.

Round 3

Reviewer 2 Report

Comments and Suggestions for Authors

The edits have been done.